# Retrospective analysis of clinical trial safety data for pembrolizumab reveals the effect of co-occurring infections on immune-related adverse events

Tigran Makunts[1]*, Keith Burkhart[2], Ruben Abagyan[1], Peter Lee[2]

1 Skaggs School of Pharmacy and Pharmaceutical Sciences, University of California San Diego, La Jolla, California, United States of America, 2 Office of Clinical Pharmacology, Center for Drug Evaluation and Research, US Food and Drug Administration, Silver Spring, Maryland, United States of America

* tmakunts@health.ucsd.edu

**Data Availability Statement:** The raw data used in the analysis cannot be made publicly available due to government regulatory and legal restrictions. The raw data sets used for the submission are

## Abstract

Biologics targeting PD-1, PD-L1, and CTLA-4 immune checkpoint proteins have been used in a variety of tumor types including small and non-small cell lung cancers, melanoma, and renal cell carcinoma. Their anti-tumor activity is achieved through amplifying components of the patient's own immune system to target immune response evading cancer cells. However, this unique mechanism of action causes a range of immune related adverse events, irAEs, that affect multiple physiological systems in the body. These irAEs, depending on severity, often cause suspension or discontinuation of therapy and, in rare cases, may lead to fatal outcomes. In this study we focused on pembrolizumab, a PD-1 inhibitor currently approved for multiple types of cancer. We analyzed over ten thousand adverse event reports from Keynote clinical trials of pembrolizumab for various cancer indications with or without co-occurring infections, and observed a statistically significant 80% increase in the risk of developing an irAE in subjects with infections.

## Introduction

The field of cancer immunotherapy has seen continuous growth and appreciation following successful efficacy trials of various targeted immune checkpoint inhibitors (ICIs). The immune system can detect and target cancer cells, however, these cells have the capacity to evolve to evade the immune system by suppressing T-cell activation [1, 2]. Multiple immune-checkpoints can modulate the T-cell response. The first ICI antibody, ipilimumab, targeted the cytotoxic T-lymphocyte–associated antigen 4 (CTLA-4)) [3]. Programmed cell death protein 1 (PD-1) receptor antibodies, pembrolizumab and nivolumab were approved in 2014, and cemiplimab in 2018 [4]. The use of the checkpoint inhibitors has been linked to significant immune-related adverse events (irAEs) [5, 6] affecting multiple organ systems and leading to irAEs such as colitis, pancreatitis, hepatitis, thyroiditis, hypophysitis, and rare but potentially fatal toxicities such as myocarditis.

allowed to be analyzed by regulatory entity. However, there is a repository for all the cases used in our analysis. The authors were not sole named individuals for ensuring data access. The adverse event cases used in the approval package, i.e. cases used in the current study, are uploaded to https://www.elsevier.com/solutions/pharmapendium-clinical-data (Critical data for comprehensive drug safety and efficacy risk assessment). The search criteria are listed in detail in the data preparation and cohort selection section. There are no special privileges necessary to access PharmaPendium other than registration and dues.

**Funding:** The authors received no specific funding for this work.

**Competing interests:** The authors have declared that no competing interests exist.

In this study we take a closer look at the irAEs associated with pembrolizumab (Keytruda). Pembrolizumab was first approved in 2014 for advanced melanoma [7], followed by over a dozen consequent approvals including non-small cell lung cancer [8], renal cell carcinoma [9], head and neck squamous cell carcinoma [10], cervical cancer and others [11, 12]. Most common adverse events of pembrolizumab as a single agent, reported in more than 20% of patients include musculoskeletal pain, fatigue, pruritus, rash, pyrexia, decreased appetite, nausea and diarrhea. Common irAEs listed on the package insert include pneumonitis, colitis, hepatitis, endocrinopathies, and nephritis. IrAEs, depending on toxicity type and Common Terminology Criteria for Adverse Events (CTCAE) [13, 14] grade, irAEs may cause suspension of treatment, permanent discontinuation, and death [15].

The systemic toxic effects of irAEs warrant further research into these adverse events. In this study we analyzed pooled data from multiple pembrolizumab clinical trial safety data submissions for various cancer types, to identify contributing factors affecting irAEs occurrence.

## Methods

### Data preparation and cohort selection

Electronic data from new molecular entity (NME) and non-NME submissions were mined using both, the integrated summaries of safety (ISS) reports, and the clinical safety summaries for pembrolizumab [16] Biologic License Application (BLA) 125514. The Adverse Event Analysis Data Set (ADAE) contained 10,023 reports which included chemotherapy, and various pembrolizumab monotherapy doses and administration schedules. Studies where subjects were administered the following doses were selected into the analysis data set (having a total of 5,732 evaluable patients): 10mg/kg every two weeks,10mg/kg every three weeks, and 200mg flat dose every three weeks. The latter dose is the current labeled recommendation [11] for adults in all cancer indications. Pembrolizumab monotherapy reports (as the only treatment for cancer) were included into the study cohort, and the chemotherapy reports were included into the control cohort. Cases with Pembrolizumab + chemotherapy and pembrolizumab 2mg/kg cases (pediatric dose) were excluded from the analysis.

Additionally, analysis dataset subject level (ADSL), analysis datasets of concomitant medications (ADCM), and medical history (MH) were used for the analysis of demographic factors and possible comorbidity and co-medication confounding effects on irAEs (Fig 1). The following parameters, previously associated with increased risk of autoimmune disease, were used in the analysis for possible confounding effects: sex [17], ethnicity [18, 19] age [20], cancer type [21, 22], co-occurring autoimmune disease [23–25], and medications associated with autoimmune AEs [26–29].

### Measured outcome(s)

The following adverse event higher level term (AEHLT MedDRA [30] hierarchy) groups were clustered together as the *irAE* outcome of interest: Colitis (excl infective), gastrointestinal inflammatory conditions, hepatocellular damage and hepatitis NEC, acute and chronic pancreatitis, acute and chronic thyroiditis, anterior pituitary hypofunction (hypophysitis), nephritis NEC, lower respiratory tract inflammatory and immunologic conditions, autoimmune disorders NEC, hemolytic immune anemias, immune and associated conditions (graft versus host disease, cytokine release), autoinflammatory diseases, and noninfectious myocarditis. The irAEs based on AEHLT designations were pooled together under the irAE umbrella term to allow for sufficient number of reports and reasonable 95% confidence intervals. The limited number of reports with organ specific irAE terms prevented any higher resolution analysis.

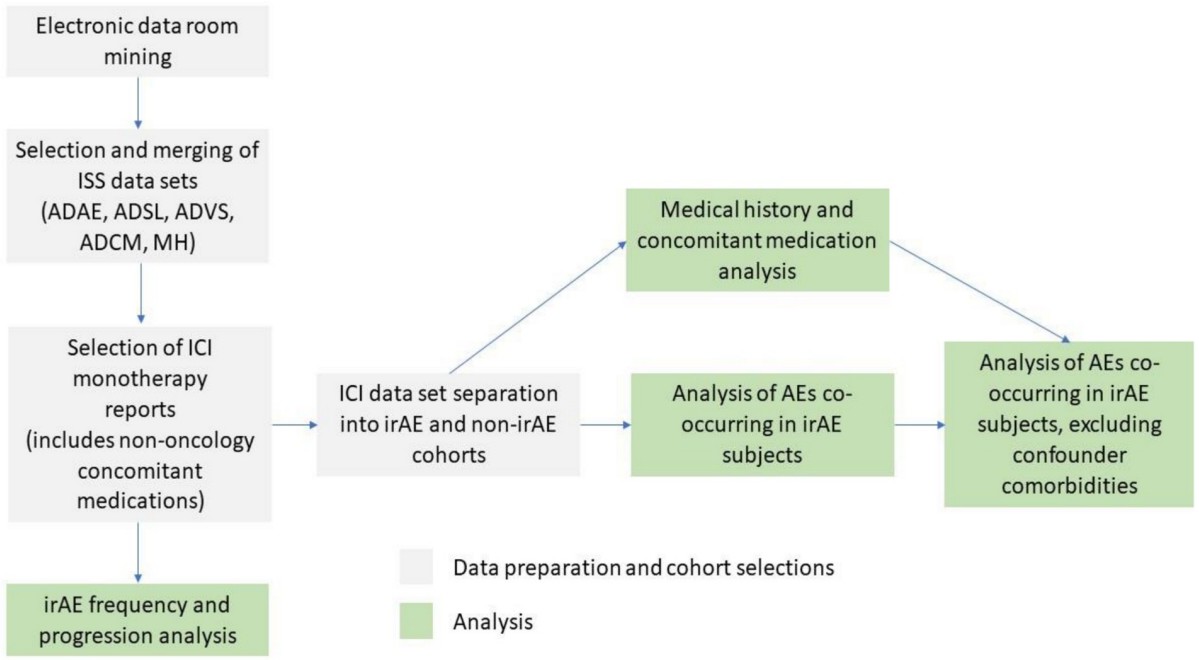

**Fig 1. Data and cohort selection, study analysis plan.**

The clustered *irAEs* were analyzed for frequency, toxicity grades, progression throughout the treatment duration, and association with other adverse events (AEs).

## Results

### Immune related adverse events

Nearly 96% of subjects experienced a Treatment Emergent Adverse Events (TEAE) of any toxicity grade during the treatment. When AEs of interest (see *irAE* in the methods section) were clustered into irAE group at AEHLT level, 9.3% of the subjects administered ICIs reported at least one irAE by the end of the study compared to 3.9% in the chemotherapy group (Table 1).

**Table 1.**

| irAE toxicity grade | % Pembrolizumab subjects with *irAEs*, n = 5,537 | % Chemotherapy subjects with *irAEs*, n = 2,108 |
|---|---|---|
| **All CTCAE grades 1–5** | **9.3** | **3.9** |
| Grade 1 | 1.7 | 0.5 |
| Grade 2 | 3.2 | 1.2 |
| Grade 3 | 3.7 | 1.9 |
| Grade 4 | 0.4 | 0.3 |
| Grade 5 | 0.3 | 0.0 |

irAEs in Pembrolizumab and Chemotherapy reports. All immune related adverse event frequencies in subjects administered pembrolizumab compared to subjects administered chemotherapy, stratified by CTCAE toxicity grade.

**Table 2.**

| irAE toxicity grade | % Pembrolizumab subjects with irAEs and infections, n = 2,528 | % Pembrolizumab subjects with irAEs without infections, n = 3,009 |
|---|---|---|
| All CTCAE grades (1–5) | 11.80 | 9.64 |
| Grade 1 | 2.49 | 1.78 |
| Grade 2 | 3.68 | 3.26 |
| Grade 3 | 4.75 | 3.86 |
| Grade 4 | 0.55 | 0.45 |
| Grade 5 | 0.32 | 0.29 |

irAEs in subjects with and without infections.

### irAEs and associated adverse events

The pembrolizumab cohort was further split into irAE and non-irAE sub-cohorts to evaluate the potential association of co-occurring irAEs. Interestingly, co-occurring infections were associated with increased frequency and severity of irAEs. The associated *infection* events included the following adverse event higher level group terms (AEHLGT): 1) Fungal infectious disorders, 2) viral infectious disorders, 3) bacterial infectious disorders, 4) mycobacterial infectious disorders, 5) infections–pathogen unspecified, 6) protozoan infectious disorders.

The statistical significance of the risk of developing an irAE with and without infections was estimated by the odds ratio (OR) value of 1.62 and 95% confidence intervals (95%CI) 1.35–1.95, $p < 0.0001$ (Table 2, Figs 2 and 3).

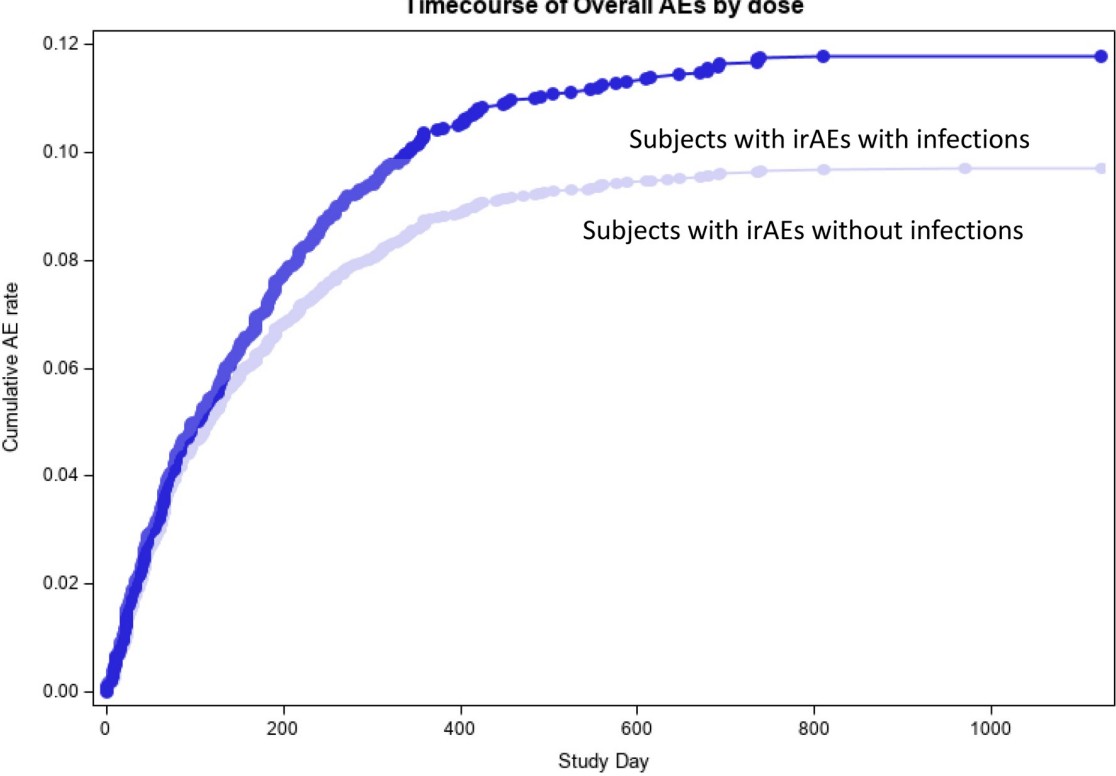

**Fig 2. Progression of irAEs throughout the treatment period in subjects with and without infections.**

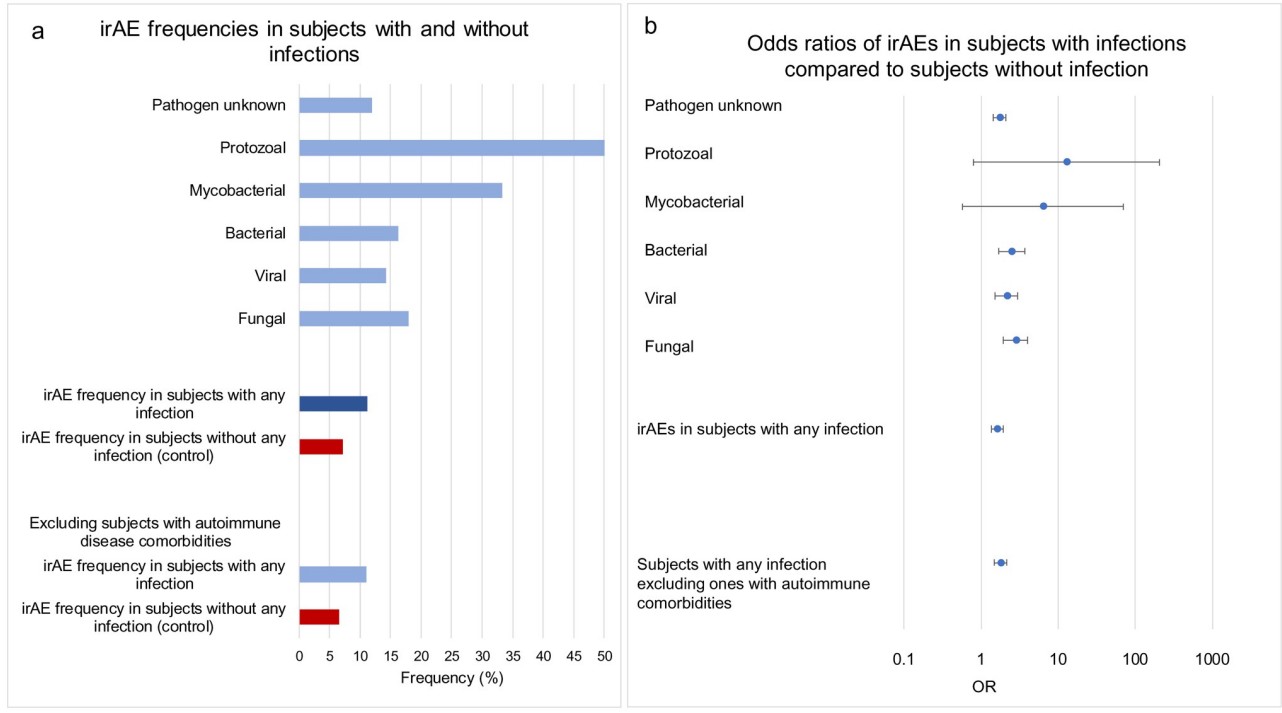

**Fig 3.** a) Frequencies or irAE in cohorts with and without infections: Infections-pathogen unknown (n = 2,052), protozoal infectious disorders (n = 2), mycobacterial infectious disorders (n = 3), bacterial infectious disorders (n = 216), viral infectious disorders (n = 336), fungal infectious disorders (n = 229), irAEs in subjects with any infection (n = 2,528), subjects with infection preceding irAE (n = 2,431), irAEs in subjects without any infection (control) (n = 3,009), subjects with any infection excluding ones with autoimmune comorbidities (n = 2,524), Subjects without any infection (control) excluding ones with autoimmune comorbidities (n = 2,984). b) Odds ratios of irAEs in subjects with infections compared to subjects without infections. X-axis presented in logarithmic scale.

## Comorbidities, demographics, and concomitant medications

Analysis of demographic parameters did not show a clinically significant difference between irAE-infections and irAE-non-infections groups (Table 3). Additionally, there were no clinically significant differences in concomitant medications associated with irAEs between the groups (Table 4). Corticosteroid administration was analyzed separately as a possible confounder for infection risk. The difference between the groups was not clinically significant either (Table 4).

The non-infection group had a nearly ten-fold higher rate to autoimmune comorbidities (11.0% vs 1.4%) (Table 4). Interestingly, when the irAE/infection association was re-analyzed excluding subjects with autoimmune comorbidities the risk of irAE increased to from 62% to 80% (OR 1.80 [1.48, 5.99]. p<0.0001) (Figs 3 and 4).

**Infection as an irAE contributing factor.** The association between irAEs and infections was statistically significant. However, due to the complexity of the data, it remained unknown whether an infection could predict an irAE. The analyzed studies included study/treatment durations ranging from 283 to 1124 days. In such a prolonged duration, many subjects had multiple occurrences of irAEs and infections, often intermittent. For this reason, time to event analysis was performed for the first infection compared to the first irAE occurrence (Table 5, Fig 5). There was a wide variety between infection and irAE start dates, with some overlap between the two, not allowing for a definitive conclusion of one predicting the other. Although

**Table 3.**

| Demographics | irAE with infection (n = 284) | irAE without infection (n = 218) |
|---|---|---|
| Mean age, years (SD) | 61.1 (14.0) | 61.3 (13.0) |
| Median age, years | 64.0 | 63.0 |
| Sex n (%) | Male 193 (69.0) | Male 159 (72.9) |
| | Female 93 (32.7) | Female 59 (27.1) |
| Ethnicity n (%) | Asian 40 (14.1) | Asian 34 (15.6) |
| | African American or Black 2 (0.7) | African American or Black 6 (2.8) |
| | White 201 (70.8) | White 145 (66.5) |
| | Multiracial 5 (1.8) | Multiracial 1 (0.5) |
| | Unknown 36 (12.7) | Unknown 31 (14.2) |

Demographic parameters of subjects with irAEs with and without infections.

infections occurred prior to irAE in most cases, nearly one third of the reports did not have the AE end date reported in the dataset, further complicating the sequalae attribution.

## Discussion

The risk of experiencing an irAE increased by nearly eighty percent, if the subjects experienced an infection sometime during the treatment. However, the irAE/infection association was not linear as the there was no clear evidence of one preceding the other (Fig 5), making adjudicating causality difficult. Some irAEs occurred prior to infections, some overlapped, and some occurred after, with the trend favoring the *infection before irAE* model. However, the association was estimated to be statistically significant.

To our best knowledge this is the first large scale study analyzing concurrent AEs in subjects experiencing irAEs. Here we analyzed over 5700 drug safety reports for subjects administered pembrolizumab in controlled clinical trials for eleven different types of cancer.

There are a few cases and smaller scale studies discussing the irAE/infection association [31, 32], and some attributing the related organ damage to irAE exacerbation due to a concurrent infection [33]. Although intuitively it makes sense that an infection can affect the irAE, this association had not been quantified in large studies. However, there are many studies that link infections with autoimmune diseases (AD), which are similar to irAEs in their manifestation and physiological profile, including autoimmune thyroiditis, colitis, and lupus [34–39] where infection related T-cell autoreactivity is the main culprit. Mechanisms by which infectious agents may cause autoimmune adverse reactions include molecular mimicry, epitope spreading, bystander activation, and cryptic antigen presentation [40].

Drawing the parallel between infection/AD and infection/irAE relationships made sense, especially in pembrolizumab administered subjects, considering the biologic's mechanism of action and the physiology behind ADs and irAEs. Interestingly, in recent studies, fecal microbiota transplants helped overcome PD-1 therapy resistance in melanoma patients [41, 42] where introducing infectious agents improved ICI treatment efficacy, suggesting the infectious agents' influence on the ICI related immune activation.

In summary, we observed a statistically significant association of co-occurring infections with immune related adverse events in pembrolizumab treated cancer patients.

### Study limitations

As with any other association study the causality between infections and irAE was not clinically adjudicated. However, the use of high-quality safety reports from controlled clinical trials

**Table 4.**

| Concomitant medications | irAE with infection cohort n of subjects irAE related concomitant medications out of 284 | irAE without infection cohort n of subjects irAE related concomitant medications out of 218 |
| --- | --- | --- |
| **irAE associated medications** | | |
| Isoniazid | 1 | 0 |
| Methimazole | 0 | 1 |
| Metoprolol | 1 | 1 |
| Hydrochlorothiazide | 1 | 1 |
| Atorvastatin | 1 | 0 |
| Fluorouracil | 1 | 2 |
| **Total unique subjects, n (100*n/n-irAE[%])** | *5 (1.76)* | *5 (2.29)* |
| **Corticosteroids used to alleviate irAEs** | | |
| Prednisone | 16 | 11 |
| Prednisolone | 12 | 9 |
| Methylprednisolone | 9 | 10 |
| Prednisone, methylprednisolone | 12 | 3 |
| Prednisolone, methylprednisolone | 8 | 4 |
| Prednisone, methylprednisolone, dexamethasone | 2 | 4 |
| Methylprednisolone, dexamethasone | 1 | 1 |
| Prednisolone, dexamethasone | 2 | 3 |
| Prednisolone betamethasone | 1 | 0 |
| Prednisolone, methylprednisolone,dexamethasone | 1 | 1 |
| Prednisolone, prednisone | 1 | 0 |
| Betamethasone, prednisolone, methylprednisolone | 1 | 2 |
| Prednisone, prednisolone, dexamethasone | 0 | 1 |
| *Dexamethasone** | 4 | 8 |
| *Betamethasone** | 3 | 1 |
| **Unique subjects, n (100*n/n-irAE[%])** | *73 (25.7)* | *58 (26.6)* |
| *Dexamethasone and betamethasone are not recommended by NCCN ICI irAE management guidelines but were included due to potential to increase infection risk. | | |

| Type of cancer, n (100*n/n-irAE[%]) | irAE with infection cohort n of subjects with irAE out of 284, n (%) | irAE without infection cohort n of subjects irAE out of 218, n (%) |
| --- | --- | --- |
| Bladder | 32 (11.2) | 21 (9.6) |
| Cervical | 2 (0.7) | 2 (0.9) |
| CRC | 4 (1.4) | 1 (0.5) |
| Gastric | 4 (1.4) | 16 (7.3) |
| HCC | 1 (0.4) | 3 (1.4) |
| HL | 12 (4.5) | 8 (3.7) |
| HNSCC | 31 (10.9) | 28 (12.8) |
| Melanoma | 91 (32.0) | 67 (30.7) |
| MLBCL | 2 (0.2) | 2 (0.9) |
| NSCLC | 92 (32.4) | 58 (26.6) |
| RCC | 9 (3.2) | 8 (3.7) |
| Unknown | 4 (1.4) | 4 (1.9) |

| irAE Subjects with related comorbidities | irAE with infection cohort n of subjects with autoimmune comorbidities out of 284 | irAE without infection cohort n of subjects with autoimmune comorbidities out of 218 |
| --- | --- | --- |
| Systemic Lupus Erythematosus | 0 | 1 |

*(Continued)*

**Table 4.** (Continued)

| Rheumatoid arthritis (arthropathies) | 1 | 3 |
|---|---|---|
| Psoriasis/psoriatic arthritis | 0 | 4 |
| Inflammatory bowel disease/IBS/UC | 1 | 5 |
| Addison's disease | 0 | 1 |
| Grave's disease/hyperthyroidism | 0 | 5 |
| Hashimoto's thyroiditis | 2 | 3 |
| Myasthenia gravis/ Lambert-Eaton's | 0 | 1 |
| Autoimmune vasculitis/Behcet's | 0 | 1 |
| Celiac disease | 0 | 1 |
| **Unique subjects, n (100\*n/n-irAE[%])** | **4 (1.4)** | **24 (11.0)** |

Medical History and Concomitant medications in irAE subjects with and without infections.

provides a strong signal that may be clinically significant. Although we analyzed concomitant medications and medical history for possible confounding effects, it needs to be noted that dietary supplements, over-the-counter medications, and even prescriptions by different providers often go underreported by patients. Additionally, minor infections such as upper respiratory

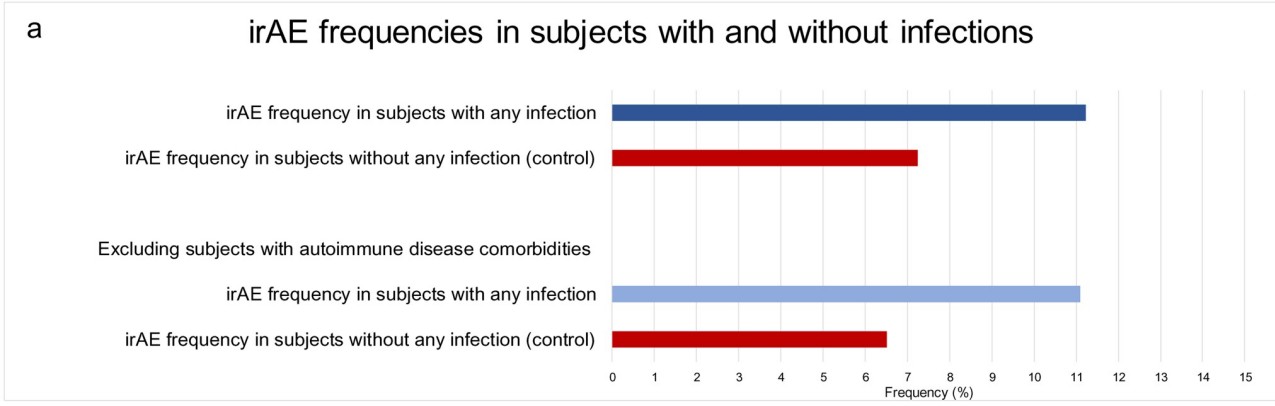

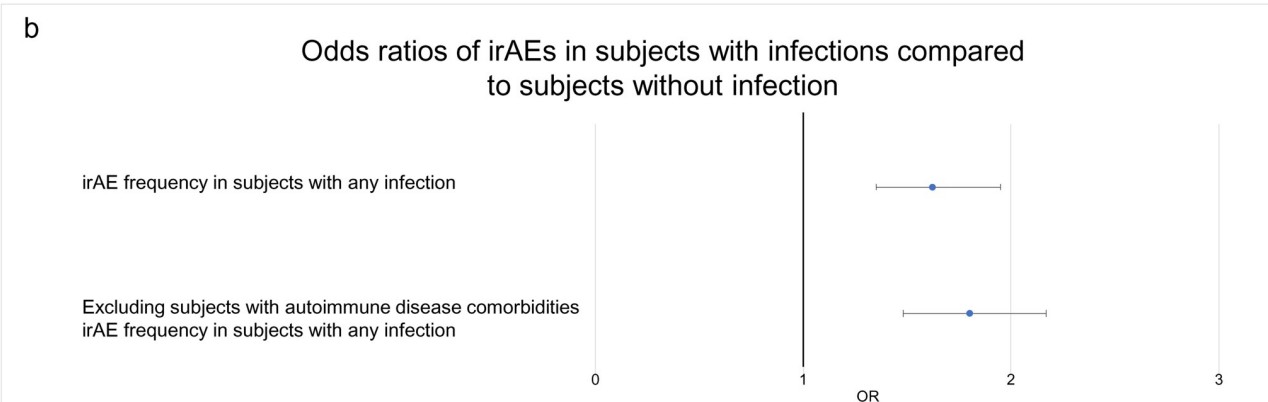

**Fig 4.** a) Frequencies or irAE in cohorts with and without infections: irAEs in subjects with any infection (n = 2,528), subjects with infection preceding irAE (n = 2,431), irAEs in subjects without any infection (control) (n = 3,009), subjects with any infection excluding ones with autoimmune comorbidities (n = 2,524), Subjects without any infection (control) excluding ones with autoimmune comorbidities (n = 2,984). b) Odds ratios of irAEs in subjects with infections compared to subjects without infections. X-axis presented in logarithmic scale.

**Table 5.**

| | Subjects with irAE and infections | | Subjects with irAE and without infections |
|---|---|---|---|
| | N = 284 | | N = 218 |
| | Time to 1st infection | Time to 1st irAE | Time to 1st irAE |
| **Mean, days (sd)** | 133.1 (140.0) | 121.3 (157.5) | 126.5 (125.2) |
| **Median, days** | 86 | 112.5 | 79 |
| **Duration, days (sd)** | 26.8 (46.2)** | 50.36** | 45.8 (60.3)** |
| **end dates for adverse events where not indicated for a large number of subjects in all the cohorts | | | |

Time to 1st infection vs time to 1st irAE.

infections may go unreported as well. This limitation may introduce some uncertainties to the analysis due to the possible autoimmunity induction by these agents [43, 44]. Unfortunately, the paucity of sufficient longitudinal data prevented further assignment of the causality to the identified significant associations.

Additionally, irAEs may have been underreported due the complexity of the diagnosis, often requiring and invasive procedure, for a definitive diagnosis, and followingly mischaracterizations of these adverse events. The study focused on only one of the currently approved antibodies in immune oncology with the goal of quantifying and establishing a signal that can

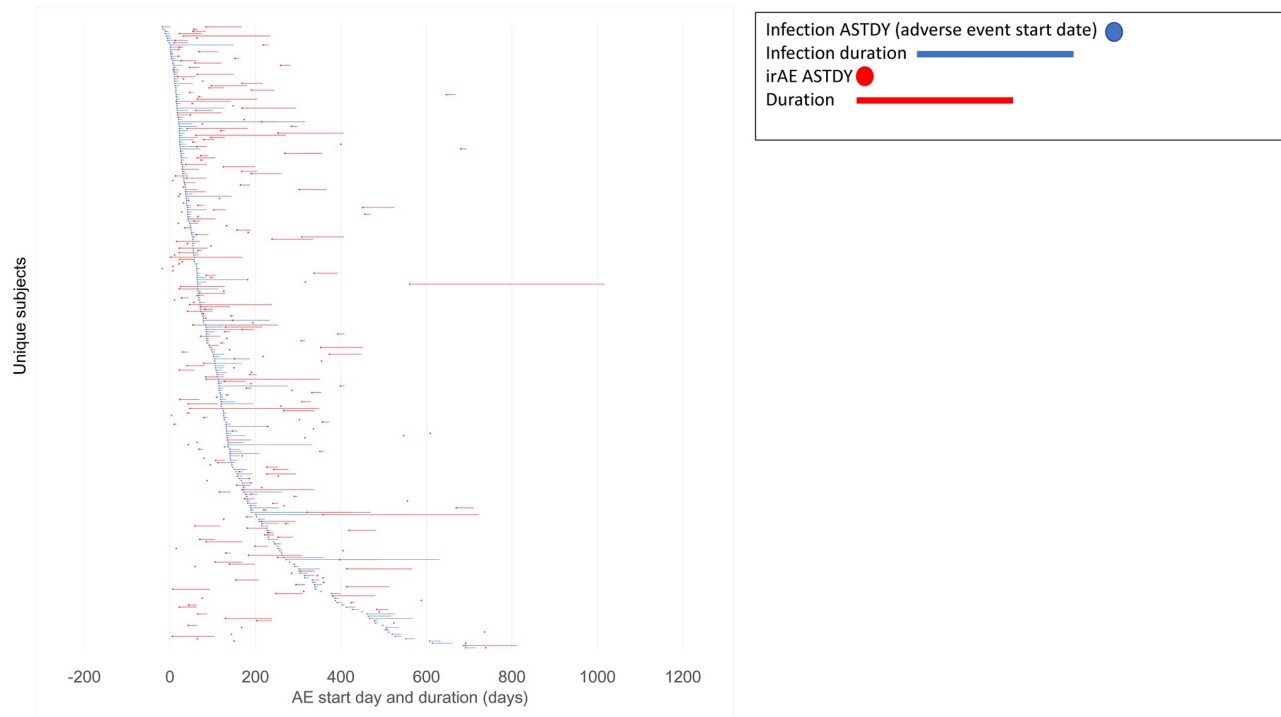

**Fig 5. First irAE and first infection time to event and duration analysis.** 1st infection (blue) vs 1st irAE (red) start date and AE duration with respect to treatment start date (day 0).

be investigated further. Future studies are needed to confirm whether the observed irAE/infection association is preserved in subjects administered other anti-PD-1, -PD-L1, and -CTLA-4 cancer immunotherapy.

## Acknowledgments

We thank Dr. Ana Szarfman for discussions on the potential for confounding and the need to show the time sequence of the assessed events. We also thank Gunjan Gugale and Constance Le for help with data management and the computer environment.

**Disclaimers**: This article reflects the views of the authors and should not be construed to represent FDA's views or policies. The results do not represent a clinical recommendation but rather illustrate a signal that needs to be further investigated in a controlled setting.

## Author Contributions

**Conceptualization:** Tigran Makunts, Keith Burkhart, Ruben Abagyan, Peter Lee.

**Data curation:** Peter Lee.

**Formal analysis:** Tigran Makunts, Peter Lee.

**Investigation:** Peter Lee.

**Methodology:** Tigran Makunts, Keith Burkhart, Ruben Abagyan, Peter Lee.

**Software:** Peter Lee.

**Supervision:** Keith Burkhart, Peter Lee.

**Validation:** Tigran Makunts.

**Visualization:** Tigran Makunts, Peter Lee.

**Writing – original draft:** Tigran Makunts, Ruben Abagyan, Peter Lee.

**Writing – review & editing:** Tigran Makunts, Keith Burkhart, Ruben Abagyan, Peter Lee.

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
