## [Decision Letter · Decision Letter 0]

17 Nov 2021

PONE-D-21-24932

Retrospective analysis of clinical trial safety data for pembrolizumab reveals the effect of co-occurring infections on immune-related adverse events

PLOS ONE

Dear Dr. Tigran Makunts,

Thank you for submitting your manuscript to PLOS ONE. After careful consideration, we feel that it has merit but does not fully meet PLOS ONE’s publication criteria as it currently stands. Therefore, we invite you to submit a revised version of the manuscript that addresses the points raised during the review process.

We look forward to receiving your revised manuscript.

Kind regards,

Yoshihiko Hirohashi, M. D., Ph. D.

Academic Editor

PLOS ONE

https://journals.plos.org/plosone/s/file?id=wjVg/PLOSOne_formatting_sample_main_body.pdf and https://journals.plos.org/plosone/s/file?id=ba62/PLOSOne_formatting_sample_title_authors_affiliations.pdf2.

2. Please clarify and explain the acronym 'ADAE' located in the manuscript at line 66.

“All authors declare no conflict of financial or non-financial interest.”

5. Please note that in order to use the direct billing option the corresponding author must be affiliated with the chosen institute. Please either amend your manuscript to change the affiliation or corresponding author, or email us at plosone@plos.org with a request to remove this option.

Reviewers' comments:

Reviewer's Responses to Questions

**Comments to the Author**

1. Is the manuscript technically sound, and do the data support the conclusions?

Reviewer #1: Partly

Reviewer #2: No

2. Has the statistical analysis been performed appropriately and rigorously? 

Reviewer #1: Yes

Reviewer #2: Yes

3. Have the authors made all data underlying the findings in their manuscript fully available?

Reviewer #1: Yes

Reviewer #2: Yes

4. Is the manuscript presented in an intelligible fashion and written in standard English?

Reviewer #1: Yes

Reviewer #2: Yes

5. Review Comments to the Author

Reviewer #1: In this paper, the authors performed retrospective analysis of clinical trial safety data and found a significant association of co-occurring infections with immune related adverse event (irAE) in pembrolizumab treated cancer patients. Although this manuscript contains some interesting findings, the authors should consider the following point.

1. Is the association between irAEs and infections characteristic of pembrolizumab? If the authors want to prove the irAE/infection association while using immune checkpoint inhibitors, they should also analyze the clinical trial data of other immune checkpoint inhibitors including nivolumab, cemiplimab, anti-PD-L1 antibodies and anti-CTLA-4 antibodies and compare those results. It is very important to clarify the association between irAEs and infections in other immune checkpoint inhibitors in this paper.

2. Is there any difference between irAEs with infection and irAEs without infection in the type of irAEs?

Reviewer #2: In this manuscript, the authors have claimed that infection may increase the risk of developing an irAE. Infection undoubtedly activate immunity. Therefore, the claim themselves would be not surprising but may confer important reference in the future studies. This reviewer has some concerns as listed below.

1. As the authors mentioned in the manuscript at around line 180 to 190, this reviewer wonders whether infection is a cause or a result of irAE. As shown in Table 5, mean days at 1st of infection delayed to that of 1st irAE. This reviewer suggests that only cases in which infection precedes irAE should be analyzed. The causal relationship between infection and irAE should be at least consistent from view of time series. If this point is not clarified, the authors' claim would be an overstatement and should not be published.

2. Is it possible that pembrolizumab exacerbate inflammatory reaction of subclinical infection? In such case, enhanced inflammation of subclinical infection by pembrolizumab is a result of irAE.

3. This reviewer is interested whether there is a difference in the clinical effect of pembrolizumab depending on the presence or absence of infection. This point would certainly improve the importance of the manuscript.

6. PLOS authors have the option to publish the peer review history of their article (what does this mean?). If published, this will include your full peer review and any attached files.

Reviewer #1: No

Reviewer #2: No

---

## [Author Response · Author response to Decision Letter 0]

26 Dec 2021

Responses to reviewer comments

Reviewer #1: 

In this paper, the authors performed retrospective analysis of clinical trial safety data and found a significant association of co-occurring infections with immune related adverse event (irAE) in pembrolizumab treated cancer patients. Although this manuscript contains some interesting findings, the authors should consider the following point.

1. Is the association between irAEs and infections characteristic of pembrolizumab? If the authors want to prove the irAE/infection association while using immune checkpoint inhibitors, they should also analyze the clinical trial data of other immune checkpoint inhibitors including nivolumab, cemiplimab, anti-PD-L1 antibodies and anti-CTLA-4 antibodies and compare those results. It is very important to clarify the association between irAEs and infections in other immune checkpoint inhibitors in this paper.

We agree with reviewer 1 that analyzing clinical trial data for other checkpoint inhibitors would be helpful. However, the study was constrained by data availability for each treatment, and the processing effort required for clinical data. We chose the antibody with the largest number of clinical trials and safety reports to ensure statistical significance and narrow 95% confidence intervals. The current study used data from every pembrolizumab clinical trial submitted to the US FDA. It took nearly 2 years of data preparation and internal government regulatory review. 

We do agree that not including every other chemotherapy in the analysis is a limitation and amended the Study limitations section with the following: 

“The study focused on only one of the currently approved antibodies in immune oncology with the goal of quantifying and establishing a signal that can be investigated further. Future studies are needed to confirm whether the observed irAE/infection association is preserved in subjects administered other anti-PD-1, -PD-L1, and -CTLA-4 cancer immunotherapy.”

2. Is there any difference between irAEs with infection and irAEs without infection in the type of irAEs?

It is definitely an interesting question. In fact, asking that question was the part of the original design of the study. Unfortunately, the paucity and ambiguity of the annotations of AE reports made it difficult to generate and any meaningful statistical analysis. The irAEs based on AEHLT designations were pooled together under the irAE umbrella term to allow for sufficient number of reports and reasonable 95% confidence intervals. 

For clarity, we improved the Measured outcomes section to explain the rationale of clustering irAEs together:

“The irAEs based on AEHLT designations were pooled together under the irAE umbrella term to allow for sufficient number of reports and reasonable 95% confidence intervals. The limited number of reports with organ specific irAE terms prevented any higher resolution analysis.”

Reviewer #2: 

In this manuscript, the authors have claimed that infection may increase the risk of developing an irAE. Infection undoubtedly activate immunity. Therefore, the claim themselves would be not surprising but may confer important reference in the future studies. This reviewer has some concerns as listed below.

Indeed, the infection and irAE association makes sense both intuitively and etiologically. However, it has never been properly quantified using a comprehensive set of reports from over a hundred controlled clinical trials for 11 types of cancers.

1. As the authors mentioned in the manuscript at around line 180 to 190, this reviewer wonders whether infection is a cause or a result of irAE. As shown in Table 5, mean days at 1st of infection delayed to that of 1st irAE. This reviewer suggests that only cases in which infection precedes irAE should be analyzed. The causal relationship between infection and irAE should be at least consistent from view of time series. If this point is not clarified, the authors' claim would be an overstatement and should not be published.

We agree that association does not necessarily indicate causation. In our study we focused on statistically significant associations that need to be further studied. Unfortunately, the paucity of sufficient longitudinal data prevented further assignment of the causality. We do mention this limitation in the results section. In the revised version we expanded the Study limitations section. See additional lines 249-251 in the Study limitations.

2. Is it possible that pembrolizumab exacerbate inflammatory reaction of subclinical infection? In such case, enhanced inflammation of subclinical infection by pembrolizumab is a result of irAE.

Thank you for the great question. A possibility of immune activation worsening certain subtypes of inflammation from infection is plausible. However, the inflammation pathways related to infections may not be the same as the immune activation through PD-1 blockade. 

In our study we demonstrate that in most cases infections precede the irAE, however the lack of AE duration end dates made the definite quantification of this observation challenging.

3. This reviewer is interested whether there is a difference in the clinical effect of pembrolizumab depending on the presence or absence of infection. This point would certainly improve the importance of the manuscript.

We agree with the reviewer that looking at efficacy as a function of existing infection would be very interesting and potentially useful. Multiple immune oncology studies have observed a positive association of severity of irAEs with efficacy. Additionally, introduction of gut microbiota concurrent with immunotherapy has shown a similar effect (see references 41 and 42). Therefore, infection may facilitate enhanced immune activation and efficacy indirectly through the above-mentioned mechanism. The study of this potential synergy was beyond the scope of our study, but definitely deserves further analysis of controlled trial data.

---

## [Editor Report · Decision Letter 1]

19 Jan 2022

Retrospective analysis of clinical trial safety data for pembrolizumab reveals the effect of co-occurring infections on immune-related adverse events

PONE-D-21-24932R1

Dear Dr. Makunts, 

We’re pleased to inform you that your manuscript has been judged scientifically suitable for publication and will be formally accepted for publication once it meets all outstanding technical requirements.

Kind regards,

Yoshihiko Hirohashi, M. D., Ph. D.

Academic Editor

PLOS ONE

Additional Editor Comments (optional):

The authors addressed concerns.
---

## [Editor Report · Acceptance letter]

27 Jan 2022

PONE-D-21-24932R1 

Retrospective analysis of clinical trial safety data for pembrolizumab reveals the effect of co-occurring infections on immune-related adverse events 

Dear Dr. Makunts:

I'm pleased to inform you that your manuscript has been deemed suitable for publication in PLOS ONE. Congratulations! Your manuscript is now with our production department. 

Kind regards, 

on behalf of

Dr. Yoshihiko Hirohashi 

Academic Editor

PLOS ONE